# Simultaneous activation of parallel sensory pathways promotes a grooming sequence in *Drosophila*

**Stefanie Hampel†, Claire E McKellar, Julie H Simpson‡\*, Andrew M Seeds†\***

Janelia Research Campus, Howard Hughes Medical Institute, Ashburn, United States

**Abstract** A central model that describes how behavioral sequences are produced features a neural architecture that readies different movements simultaneously, and a mechanism where prioritized suppression between the movements determines their sequential performance. We previously described a model whereby suppression drives a *Drosophila* grooming sequence that is induced by simultaneous activation of different sensory pathways that each elicit a distinct movement (Seeds et al., 2014). Here, we confirm this model using transgenic expression to identify and optogenetically activate sensory neurons that elicit specific grooming movements. Simultaneous activation of different sensory pathways elicits a grooming sequence that resembles the naturally induced sequence. Moreover, the sequence proceeds after the sensory excitation is terminated, indicating that a persistent trace of this excitation induces the next grooming movement once the previous one is performed. This reveals a mechanism whereby parallel sensory inputs can be integrated and stored to elicit a delayed and sequential grooming response.
DOI: https://doi.org/10.7554/eLife.28804.001

**\*For correspondence:**
julie.simpson@lifesci.ucsb.edu (JHS);
seeds.andrew@gmail.com (AMS)

**Present address:** †Institute of Neurobiology, University of Puerto Rico-Medical Sciences Campus, San Juan, Puerto Rico; ‡Department of Molecular, Cellular, and Developmental Biology, University of California, Santa Barbara, Santa Barbara, United States

**Competing interests:** The authors declare that no competing interests exist.

## Introduction

A major question about nervous system function is how different movements are assembled to form behavioral sequences. One of the primary models of sequential behavior is reminiscent of how animals select among competing behavioral choices. Behavioral competition arises in situations where different mutually exclusive behaviors are appropriate, but they must be performed one at a time (*Houghton and Hartley, 1995*; *Redgrave et al., 1999*). These conflicts can be resolved through the suppression of all but the highest priority behavior, as mollusks do to suppress their mating behavior while feeding (*Davis, 1979*; *Kupfermann and Weiss, 2001*; *Kristan, 2008*). In the case of a behavioral sequence, it is proposed that the different movements to be performed are similarly readied in parallel and in competition for output, and a suppression hierarchy determines their priority order of execution (*Lashley, 1951*; *Houghton and Hartley, 1995*; *Bullock, 2004*). Completion of the highest priority movement lifts suppression on movements of lower priority that are subsequently performed according to a new round of competition and suppression. This *parallel model* could drive behaviors across a range of complexity, from the sequential typing of letters on a keyboard in humans to the selection of which behavior to perform first in mollusks (*Houghton and Hartley, 1995*). Thus, the identification of examples of simple parallel neural architectures that drive a prioritized selection of movements may inform a broad spectrum of sequential behaviors (*Kristan, 2014*; *Jovanic et al., 2016*).

A *Drosophila melanogaster* grooming sequence provides one example of how conflicting stimuli can induce movement competition that is resolved through a suppression hierarchy. Coating the body of a fly with dust is thought to stimulate competition among different grooming movements that are each responsible for cleaning a particular body part (*Phillis et al., 1993*; *Seeds et al.,*

*2014*). We previously presented evidence that the body grooming order is determined through a mechanism where earlier movements suppress later ones (*Seeds et al., 2014*). For example, removal of dust from the eyes occurs first because eye grooming suppresses cleaning of the other body parts. From a suppression hierarchy among the different grooming movements emerges a sequence that proceeds in the order: eyes > antennae > abdomen > wings > notum. We further proposed a computational model to describe this sequence that features parallel activation of the different grooming movements by dust to induce competition, and hierarchical suppression among the movements to determine their selection order (*Seeds et al., 2014*). The parallel activation of the movements was proposed based on evidence that stimulation to each body part induces site-directed grooming responses (*Vandervorst and Ghysen, 1980*; *Corfas and Dudai, 1989*; *Seeds et al., 2014*; *Hampel et al., 2015*). Thus, the simultaneous, or parallel stimulation of sensory neurons by dust would cause different grooming movements to compete for output because only one can be performed at a time. However, it was not confirmed that simultaneous activation of sensory neurons across the body indeed elicits the same prioritized grooming response that we observed using a dust stimulus.

Here, we reveal a neural basis for parallel activation of sensory inputs for a sequential behavior by identifying sensory neurons that stimulate different grooming movements, and by testing the hypothesis that activation of these neurons in parallel elicits a prioritized grooming response. We identify transgenic expression tools for visualizing and optogenetically activating sensory neurons on the body parts that elicit specific grooming movements. This enables the simultaneous activation of sensory neurons across the body to induce competition among their respective grooming movements. As we observed by coating the bodies of flies in dust, whole-body sensory activation elicits grooming that prioritizes the head and then proceeds to the other body parts. This provides direct evidence that the grooming sequence can be induced through simultaneous activation of sensory neurons across the body. These experiments also reveal that flies have a persistent trace of the body parts that were stimulated, which results in delayed and sequential grooming of the stimulated parts. Work presented here lends neural-based evidence in favor of the parallel model of hierarchical suppression among grooming movements and provides new insights into its underlying organization.

## Results

### GAL4 lines targeting sensory neurons across the body that elicit grooming

Our initial goal was to identify GAL4 transgenic lines expressing in sensory neurons across the body, to directly test whether simultaneous activation of these neurons leads to a prioritized grooming response. As an entry point, we examined a collection of previously identified enhancer-driven GAL4 lines that express in different neuronal populations whose activation drove grooming (*Seeds et al., 2014*). Confocal microscopy imaging of the *peripheral nervous system* (PNS) expression patterns of different lines from this collection revealed three that express in sensory neurons across the body (R52A06-, R30B01-, and R81E10-GAL4; *Figure 1A–F*, R52A06-GAL4 shown as an example, *Figure 1—figure supplement 1A–D*). We classified the different sensory neuron types based on previous anatomical descriptions (*Murphey et al., 1989*; *Cole and Palka, 1982*; *Dickinson and Palka, 1987*; *Smith and Shepherd, 1996*; *Kays et al., 2014*) and found that the lines express predominantly in mechanosensory neurons (*Figure 1G*). However, R30B01-GAL4 also showed expression in chemosensory neurons (*Figure 1G*).

We next tested whether local populations of sensory neurons on specific body regions can elicit individual grooming movements when focally activated. Site-directed grooming responses have previously been investigated using tactile stimulation to particular mechanosensory bristles on the body surface of decapitated flies (*Vandervorst and Ghysen, 1980*; *Corfas and Dudai, 1989*). Here, we used optogenetic activation with Channelrhodopsin (ChR2), directing blue light via an optical fiber to particular body regions of the broad sensory GAL4 lines to activate sensory neurons on either the dorsal anterior or posterior body regions of decapitated flies (*Figure 1—figure supplement 2A,B*). Light directed to the posterior dorsal body surface elicited grooming of the wings, whereas illumination of the anterior dorsal surface elicited grooming of the notum (*Figure 1H*, *Video 1*, *Video 2*). This indicated that site-directed grooming responses can be elicited optogenetically, and that the

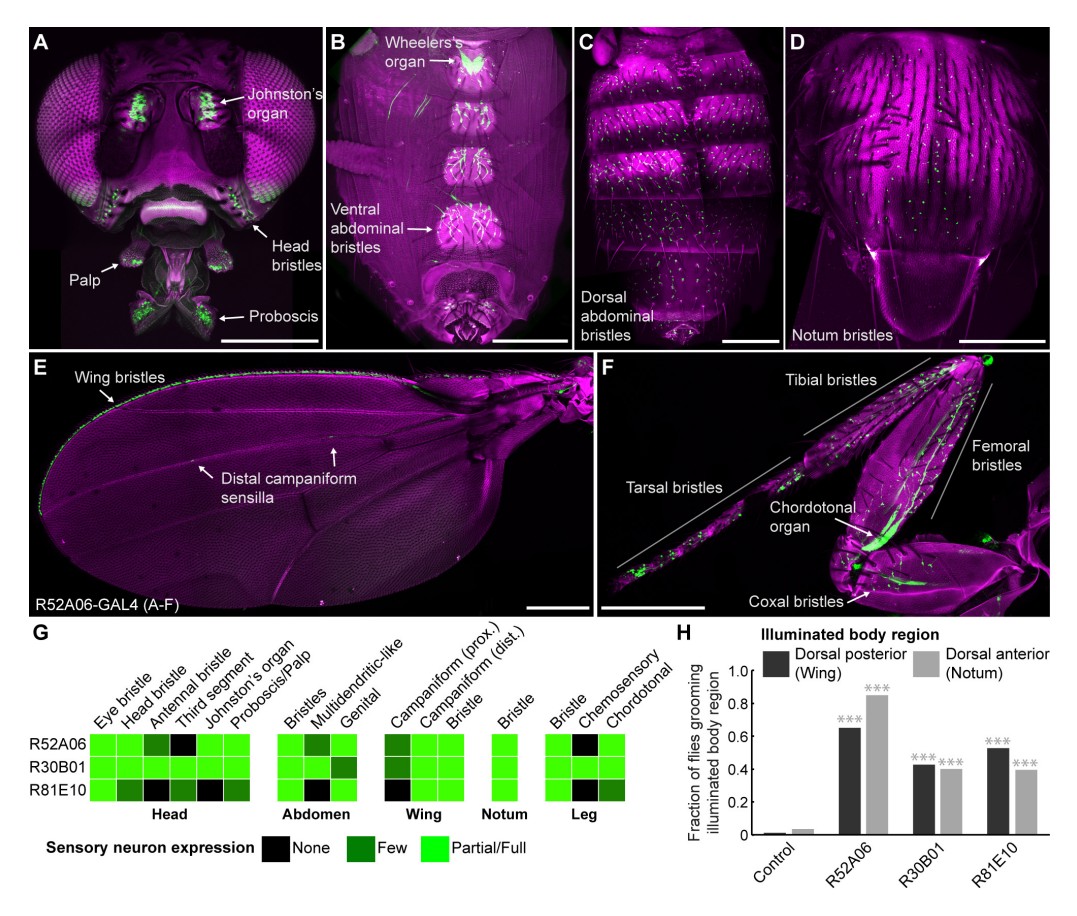

**Figure 1.** GAL4 lines expressing in sensory neurons whose activation elicits grooming. (**A–E**) Peripheral expression pattern of R52A06-GAL4 expressing *green fluorescent protein* (GFP). Confocal maximum projections are shown. Sensory neurons are in green and autofluorescence from the cuticle is in magenta. Body parts shown are: (**A**) head, (**B**) ventral abdomen, (**C**) dorsal abdomen, (**D**) notum, (**E**) wing, and (**F**) prothoracic leg. Labeled arrows indicate specific sensory classes. In (**C**) and (**D**) all GFP positive cells are bristle mechanosensory neurons. Scale bars, 250 μm. (**G**) Summary table of the expression patterns of R52A06-, R30B01-, and R81E10-GAL4 in sensory neurons on each indicated body part. (**H**) Grooming responses to optogenetic activation of sensory neurons targeted by different GAL4 lines expressing ChR2. An optical fiber connected to an LED was used to direct light to the dorsal surface of the anterior or posterior body (*Figure 1—figure supplement 2*). The fraction of flies that showed a grooming response to the blue light-illuminated body region is plotted (n ≥ 40 trials for each body part). Grey shades and labels indicate the region that was illuminated. Chi-squared test, Asterisks: p<0.0001. See *Video 1* and *Video 2* for representative examples.

DOI: https://doi.org/10.7554/eLife.28804.002

The following figure supplements are available for figure 1:

**Figure supplement 1.** Anatomy of sensory GAL4 lines.
DOI: https://doi.org/10.7554/eLife.28804.003

**Figure supplement 2.** Optogenetic illumination of sensory neurons on different body regions.
DOI: https://doi.org/10.7554/eLife.28804.004

GAL4 lines express in sensory neurons whose activation can elicit grooming movements for at least two parts of the body.

## Simultaneous excitation of sensory neurons across the body induces a grooming sequence

The GAL4 lines described above were next used to test a prediction of the model of hierarchical suppression that simultaneous activation of sensory neurons across the body elicits head grooming preferentially (*Seeds et al., 2014*). Freely moving flies of each line expressing the red light-gated neural activator CsChrimson were exposed to whole body illumination to optogenetically activate their targeted sensory neurons, and grooming responses were subsequently measured. Each of the

three GAL4 lines expresses in sensory neurons whose activation can elicit wing or notum grooming, as revealed by localized optogenetic activation (*Figure 1G,H*). Additionally, each line expresses in eye bristle mechanosensory neurons whose activation we hypothesized could elicit eye grooming, while two of the lines (R52A06- and R30B01-GAL4) also express in antennal Johnston's Organ neurons that were previously shown to elicit antennal grooming (*Hampel et al., 2015*). Although these GAL4 lines can elicit several movements from different body sensory neurons, we predicted that activating them simultaneously should elicit only the highest-priority movement, according to the hierarchical suppression model. Indeed, the simultaneous optogenetic activation of body sensory neurons targeted by each GAL4 line resulted in head rather than posterior (abdomen, wing, notum) grooming, consistent with the model of hierarchical suppression (*Figure 2A*, during red light-on period, *Figure 2—figure supplement 2A,B*).

Optogenetic activation of sensory neurons across the body also elicited a grooming sequence reminiscent of dust-induced grooming. Flies groomed their heads at the onset of a five-second red light stimulus, and then transitioned to grooming their posterior bodies during the period after the light was turned off (*Figure 2A*, *Figure 2—figure supplement 2A,B*). One trivial explanation for this sequence could be that optogenetic activation of sensory neurons on the posterior body elicited grooming with a delay, whereas there was no delay to groom with activation of the head sensory neurons. We tested for this delay to groom the posterior body using decapitated flies that no longer received a sensory drive to groom their heads. In contrast to intact flies, activation of the posterior body sensory neurons of decapitated flies elicited posterior grooming during the red light (*Figure 2B*). Thus, a delay does not explain the sequence because head and posterior grooming can be elicited on similar time scales. Instead, evidence that intact flies do not display posterior grooming with the light stimulation supports the hypothesis that it is suppressed by head grooming (discussed below). Notably, optogenetic activation of sensory neurons across the body causes flies to groom their bodies in the same order as when they were coated in dust (head > abdomen > wings > notum) (*Figure 2—figure supplement 3A,B*, *Video 3*, *Video 4*, and *Video 5*). Further, the posterior body grooming sequence continued through the minute after the cessation of the red light, while the sensory neurons were no longer activated (*Figure 2A*, green histogram). This suggests a persistent trace of posterior sensory neurons that had been activated, which allowed each movement to be elicited once the previous grooming movement terminated.

The behavior resulting from simultaneous activation of sensory neurons across the body supports a role of suppression in establishing the grooming movement hierarchy. Evidence of suppression was found when sensory neurons were reactivated during the period when flies had transitioned to posterior grooming (*Figure 2A*). The hierarchical suppression model predicts that switching the red

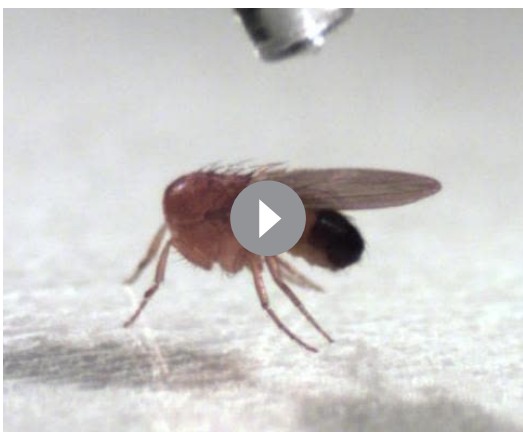

**Video 1.** Grooming response to blue light illumination of the dorsal posterior body surface of a decapitated fly expressing ChR2 in sensory neurons. ChR2 was expressed in sensory neurons across the body using R52A06-GAL4.
DOI: https://doi.org/10.7554/eLife.28804.005

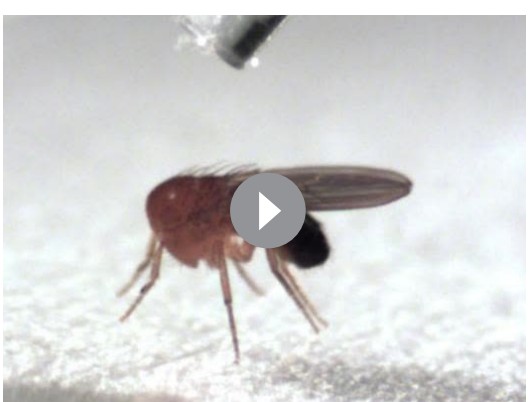

**Video 2.** Grooming response to blue light illumination of the dorsal anterior body surface of a decapitated fly expressing ChR2 in sensory neurons. ChR2 was expressed in sensory neurons across the body using R52A06-GAL4.
DOI: https://doi.org/10.7554/eLife.28804.006

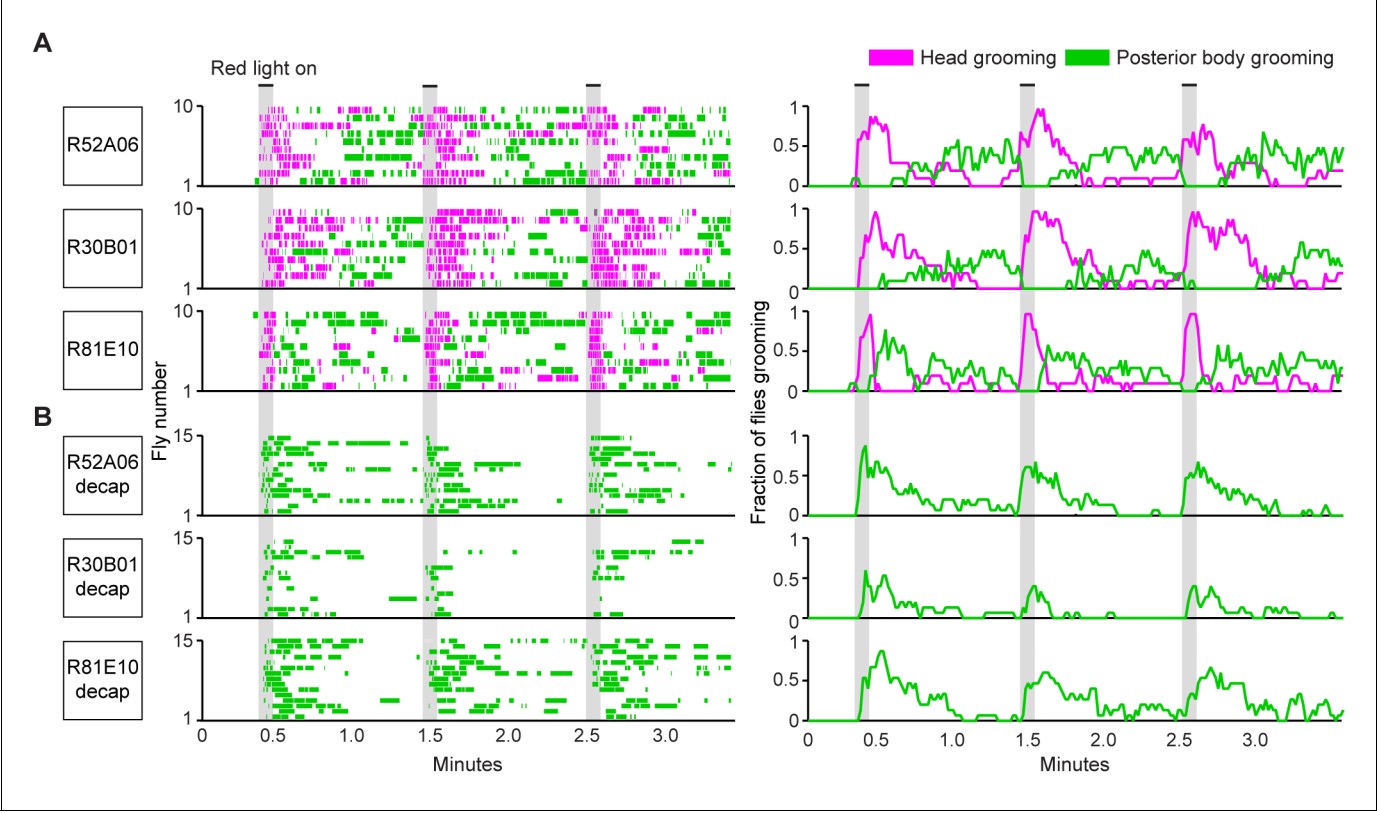

**Figure 2.** Simultaneous optogenetic activation of body sensory neurons elicits sequential grooming. (A–B) Head (magenta) or posterior body grooming movements (green) elicited with red light-illumination of R52A06-, R30B01-, and R81E10-GAL4 flies expressing CsChrimson. The movements are mutually exclusive. Ethograms of ten individual flies are stacked for each line (left). Histograms show the fraction of flies that were performing specific grooming movements within one-second time bins (right). Gray bars indicate five second presentations of red light. (A) Grooming movements performed by intact flies. (B) Grooming movements performed by decapitated flies. See *Video 3*, *Video 4*, and *Video 5* for representative examples. Red light illumination of control flies did not elicit grooming (*Figure 2—figure supplement 1*).
DOI: https://doi.org/10.7554/eLife.28804.007

The following figure supplements are available for figure 2:

**Figure supplement 1.** Illumination of control flies does not elicit grooming.
DOI: https://doi.org/10.7554/eLife.28804.008
**Figure supplement 2.** Simultaneous optogenetic activation of body sensory neurons elicits a prioritized head grooming response.
DOI: https://doi.org/10.7554/eLife.28804.009
**Figure supplement 3.** Simultaneous optogenetic activation of body sensory neurons elicits a grooming sequence.
DOI: https://doi.org/10.7554/eLife.28804.010
**Figure supplement 4.** Simultaneous optogenetic activation of body sensory neurons elicits eye grooming and terminates ongoing posterior grooming.
DOI: https://doi.org/10.7554/eLife.28804.011

light back on during this period to reactivate sensory neurons across the body would result in head grooming, coupled with the termination of ongoing posterior grooming. Indeed, in cases where flies were engaged in posterior grooming, delivery of the next red light stimulus caused flies to terminate posterior grooming and switch to grooming their heads. This is seen in *Figure 2A* (histogram plots on right, green traces) where the fraction of flies grooming their posterior bodies drops to zero at the onset of the red light (also shown in *Figure 2—figure supplement 2B*). This termination of posterior grooming was still observed when we shortened the time between light stimuli and only examined trials where flies were grooming at the moment the next stimulus was delivered (*Figure 2—figure supplement 4*). Thus, we find optogentic evidence consistent with the hypothesis that the grooming sequence is driven by a hierarchical suppression mechanism, as was revealed from experiments using dust as a natural stimulus (*Seeds et al., 2014*).

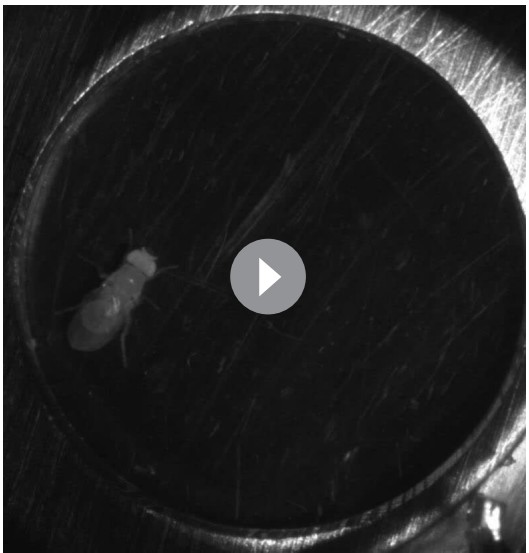

**Video 3.** Grooming in response to the simultaneous optogenetic activation of sensory neurons across the body (R52A06-GAL4). CsChrimson was expressed in sensory neurons using R52A06-GAL4. The infrared light in the bottom right corner indicates when the red light was on to activate the sensory neurons.
DOI: https://doi.org/10.7554/eLife.28804.012

## Identification of mechanosensory neurons that elicit specific grooming movements

We next sought to test whether the hierarchy of grooming movements could be observed with competing activation of defined sets of sensory neurons that elicit distinct movements. We first acquired transgenic lines for manipulating sensory neurons on specific body parts. Eye grooming is the most hierarchically superior, and is thus elicited first in competition with other grooming movements (*Seeds et al., 2014*). Based on previous work implicating the interommatidial bristle mechanosensory neurons in eye grooming in the praying mantis and cricket (*Honegger, 1977*; *Honegger et al., 1979*; *Zack and Bacon, 1981*), we found that these neurons elicit eye grooming in *Drosophila*. A search through an image database of brain expression patterns from the Vienna *Drosophila* collection identified a LexA line (VT17251-LexA) that expressed exclusively in the interommatidial bristle mechanosensory neurons. The hundreds of bristles on the compound eyes each contains the dendrite of a sensory neuron, which also projects an axon into an afferent tract that enters the CNS in the *subesophageal zone* (SEZ)

(*Figure 3A,B*). In contrast to the praying mantis and cricket, the fly eye bristle afferents project only to the SEZ, and not also the prothoracic neuromeres (*Figure 3B*). We tested whether activation of eye bristle mechanosensory neurons would elicit grooming by expressing CsChrimson using

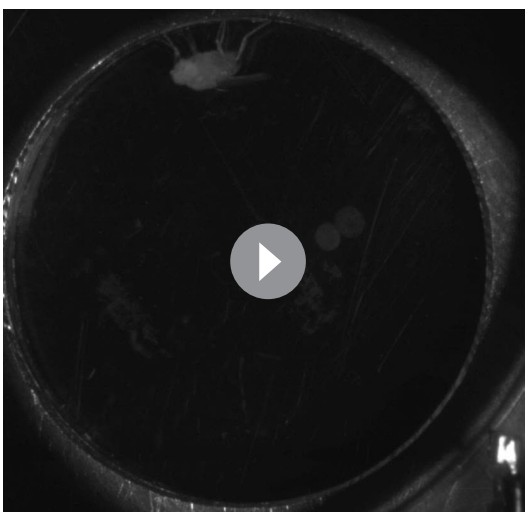

**Video 4.** Grooming in response to the simultaneous optogenetic activation of sensory neurons across the body (R30B01-GAL4). CsChrimson was expressed in sensory neurons using R30B01-GAL4. The infrared light in the bottom right corner indicates when the red light was on to activate the sensory neurons.
DOI: https://doi.org/10.7554/eLife.28804.013

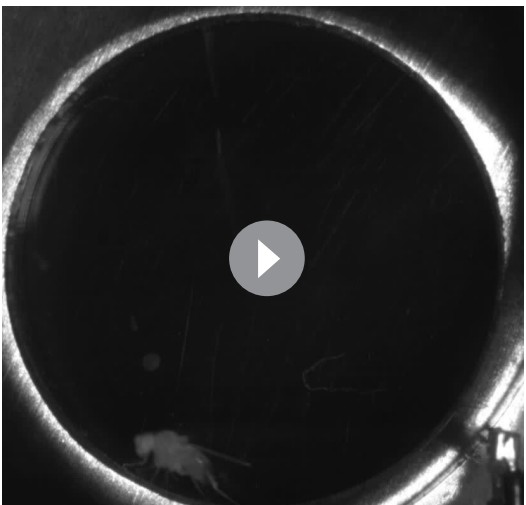

**Video 5.** Grooming in response to the simultaneous optogenetic activation of sensory neurons across the body (R81E10-GAL4). CsChrimson was expressed in sensory neurons using R81E10-GAL4. The infrared light in the bottom right corner indicates when the red light was on to activate the sensory neurons.
DOI: https://doi.org/10.7554/eLife.28804.014

VT17251-LexA and exposing flies to red light. Indeed, optogenetic activation of the eye bristle mechanosensory neurons elicited eye grooming (*Figure 3C*).

We next acquired a transgenic driver line for manipulating sensory neurons that could elicit wing grooming, which is lower in the hierarchy than eye grooming. From our previous screen (*Seeds et al., 2014*), we identified a GAL4 line that expresses in neurons whose activation could elicit wing grooming and showed expression in sensory neurons on the wings (*Figure 4—figure supplement 1A*, R31H10-GAL4, behavioral data not shown). However, the identities of those sensory neurons were obscured by expression in other cells (*Figure 4—figure supplement 1B*). Therefore, we used the intersectional *Split GAL4* (spGAL4) technique to restrict expression to only the sensory neurons (*Luan et al., 2006*; *Pfeiffer et al., 2010*). spGAL4-mediated expression occurs only when the two GAL4 domains, the GAL4 *DNA binding domain* (DBD) and the *transcriptional activation domain* (AD), are expressed in the same cells. We generated spGAL4 flies that were anticipated to target the wing sensory neurons by expressing the DBD in the pattern of R31H10-GAL4 and the AD in the pattern of R30B01-GAL4 (*Figure 4—figure supplement 1B,C*).

The R30B01-AD ∩ R31H10-DBD combination expresses in two main types of mechanosensory neurons on the wings and halteres (*Figure 4A–E*). The first type includes campaniform sensilla, which are dome-shaped structures on the fly cuticle that are each innervated by a mechanosensory neuron that responds to deformations of the cuticle (*Dickinson and Palka, 1987*). Campaniform sensilla on the proximal part of the wing are largely clustered in fields, whereas individual sensilla are found along the distal wing (*Palka et al., 1979*; *Cole and Palka, 1982*; *Palka et al., 1986*; *Dickinson and Palka, 1987*). R30B01-AD ∩ R31H10-DBD flies show a sparse labeling of neurons in the proximal fields (5 to 10 out of ~77 neurons (median = 6.5), *Figure 4A,B*, white asterisks), and expression in the majority of the distal campaniform sensilla (5 to 6 out of 8 neurons (median = 5), *Figure 4A,C*, yellow asterisks). The spGAL4 line also expresses in campaniform sensilla on the halteres (7 to 10 out of ~139 neurons, *Figure 4E*). The other type of sensory neurons targeted by R30B01-AD ∩ R31H10-DBD are mechanosensory bristle neurons on the distal wing (expression in 3–5 out of ~221 neurons (median = 3.5), *Figure 4A,D*, white arrowheads) (*Hartenstein and Posakony, 1989*). These different neurons on the wings and halteres send projections to the *ventral nervous system* (VNS), where they follow diverse paths locally, with some further ascending to the SEZ in the brain (*Figure 4F*). The ascending afferents are likely to be from campaniform sensilla on the halteres and proximal wings, whereas afferents that remain in the VNS are likely to be from wing mechanosensory bristle neurons and distal campaniform sensilla (*Palka et al., 1979*; *Ghysen, 1980*; *Dickinson and Palka, 1987*).

Optogenetic activation of the neurons targeted by R30B01-AD ∩ R31H10-DBD expressing CsChrimson elicited wing but not haltere grooming (*Figure 4G*). The parsimonious explanation for this result is that the grooming was elicited by sensory neurons on the wing. However, because the line also expresses in haltere campaniform sensilla, we cannot rule out their involvement in the behavior. Nevertheless, the spGAL4 driver affords access to sensory neurons for independent control of wing grooming.

## Competition between eye and wing sensory neurons elicits prioritized grooming

The hierarchical associations between eye and wing grooming were next examined by activating their respective sensory pathways. We first compared the individual grooming responses to acute activation of either the eye bristle mechanosensory neurons or the wing/haltere sensory neurons. Flies were exposed to five-second pulses of red light, followed by rest periods with no light. Activation of the eye bristle mechanosensory neurons elicited eye grooming during the period when the red light was on that decayed when it turned off (*Figure 5A*, top, magenta, *Video 6*). In contrast, activation of the wing/haltere sensory neurons elicited wing grooming with the red light that persisted after light cessation (*Figure 5A*, middle, green, *Video 7*). Importantly, activation of either the eye bristle mechanosensory neurons or the wing sensory neurons alone did not elicit the other corresponding grooming movement, or an anterior-to-posterior grooming sequence. Thus, activation of these specific sensory types only elicits grooming of its corresponding body part.

We next tested whether activation of the eye bristle mechanosensory neurons and wing/haltere sensory neurons at the same time would elicit a prioritized eye grooming response, as is predicted by the model. For this experiment, we identified a spGAL4 combination (R31H10-AD ∩ R34E03-DBD) that expressed both in the eye bristle mechanosensory neurons and the same three categories

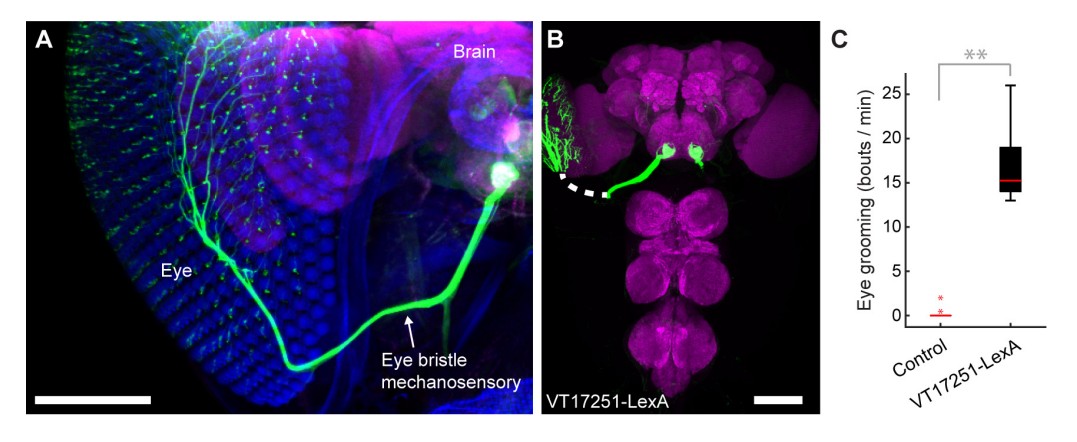

**Figure 3.** Interommatidial bristle mechanosensory neurons elicit eye grooming. (**A–B**) The expression pattern of VT17251-LexA in eye bristle mechanosensory neurons. The neurons were stained with anti-GFP (green) and the brain neuropile is stained with anti-Bruchpilot (magenta). Both images are maximum intensity projections. Scale bars, 100 μm. (**A**) Expression pattern shown in the semi-intact head. The eye and head cuticle is shown in blue. (**B**) Expression pattern in the CNS. White dashed line indicates the trajectory of eye bristle mechanosensory neuron axons found from the more intact preparations in (**A**). (**C**) Eye grooming bout rate with optogentic activation of neurons targeted by VT17251-LexA. Bottom and top of the boxes indicate the first and third quartiles respectively; median is the red line; whiskers show the upper and lower 1.5 IQR; red dots are data outliers (n = 10 for each box; asterisks show p<0.001, Kruskal-Wallis and post hoc Mann-Whitney U pairwise test).

DOI: https://doi.org/10.7554/eLife.28804.015

of sensory neurons on the wings and halteres that were expressed in the R30B01-AD ⋂ R31H10-DBD combination (*Figure 5B*). Simultaneous optogenetic activation of these defined sensory neurons elicited prioritized grooming that started with the eyes and then proceeded to the wings (*Figure 5A*, bottom, *Figure 5—figure supplement 1A,B*, *Video 8*), like what we observed with activation of sensory neurons across the body (*Figure 2*). Interestingly, the transition of eye to wing grooming occurred within the stimulus period, suggesting that eye grooming became inhibited by wing grooming during the stimulus. This may reveal a prediction of the model of hierarchical suppression that later movements in the sequence can suppress earlier ones (see Discussion). However, the prioritized suppression came from eye grooming, as any ongoing grooming of the wings terminated and all flies groomed their eyes with each red light stimulus (*Figure 5A*, bottom, *Figure 5—figure supplement 1A,B*). This experiment demonstrates the prioritization between grooming movements through direct optogenetic activation of the sensory neurons that elicit grooming of specific body parts. This strengthens the conclusion of our previous work that the sequence occurs when the grooming movements are activated in parallel and then sequentially prioritized through hierarchical suppression (*Seeds et al., 2014*).

## Discussion

The goal of this work was to test the prediction of the model of hierarchical suppression that simultaneous activation of sensory neurons on different body parts elicits a prioritized grooming response. Two lines of evidence led us to this prediction. The first was based on our previous finding that coating the body of the fly in dust elicits grooming that prioritizes head over posterior body grooming (*Seeds et al., 2014*). The second was based on data showing that local stimulation to the body surface elicits site-specific grooming responses (*Vandervorst and Ghysen, 1980*; *Corfas and Dudai, 1989*; *Seeds et al., 2014*; *Hampel et al., 2015*). Thus, we proposed that sensory neurons across the body are stimulated in parallel by dust to elicit competition among their respective grooming movements. Here, we test this by identifying transgenic driver lines for targeting and directly activating sensory neurons that elicit grooming, allowing us to bypass the dust stimulus and reveal the underlying sensory neurons. Using simultaneous optogenetic activation of sensory neurons across the body we observe the same anterior-to-posterior prioritization among the grooming movements that occurs when flies are coated in dust. This lends strong support to the hypothesis that the grooming

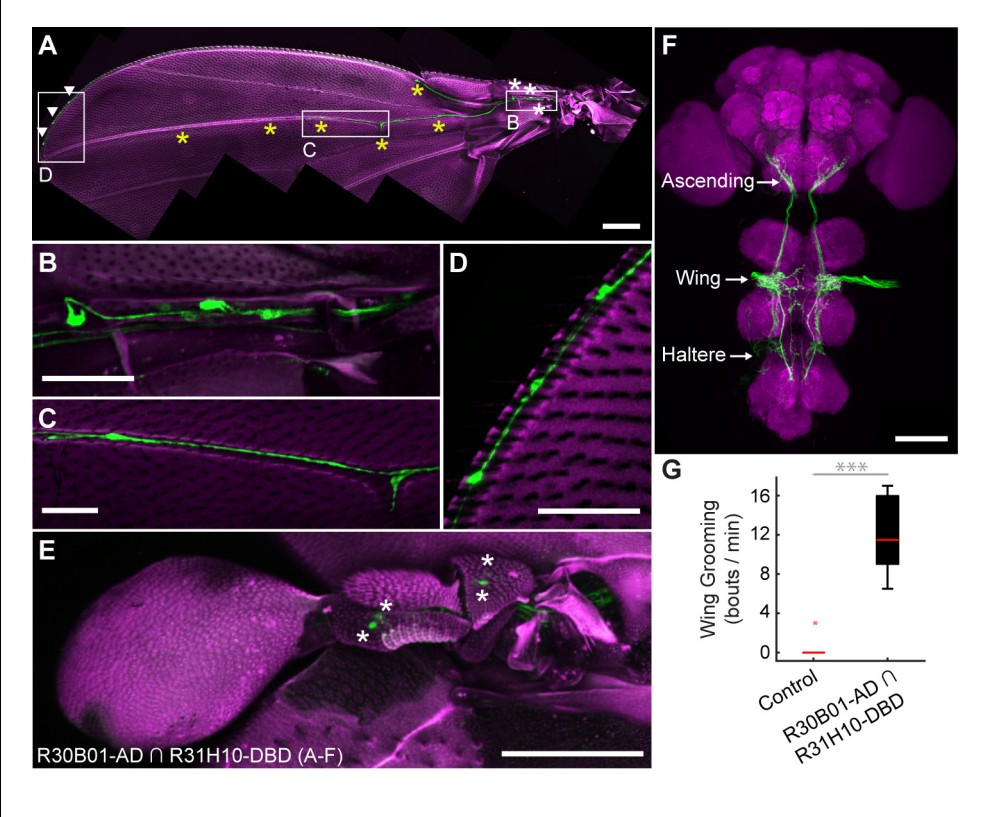

**Figure 4.** spGAL4 driver that expresses in wing and haltere sensory neurons whose activation elicits wing grooming. (**A–E**) The expression pattern of R30B01-AD ∩ R31H10-DBD in sensory neurons of the wings and halteres. Native GFP fluorescence is shown in green and autofluorescence from the cuticle is in magenta. Maximum intensity projections are shown. The proximal wing is to the right and the distal wing is to the left. (**A**) Sensory neurons on the wing. White boxes and letters indicate the regions shown in **B–D**. The different symbols indicate the sensory neuron types on the wing as proximal campaniform sensilla (white asterisks), distal campaniform sensilla (yellow asterisks), or bristle mechanosensory (white arrowheads). Scale bar, 250 µm. (**B–D**) Larger images of the regions shown in **A**. Scale bars, 50 µm. Shown are the proximal campaniform sensilla (**B**), distal campaniform sensilla (**C**), and bristle mechanosensory neurons (**D**). (**E**) Expression in the haltere campaniform sensilla (asterisks). Scale bar, 100 µm (**F**) CNS expression visualized by co-stain with anti-GFP (green) and anti-Bruchpilot (magenta). Arrows indicate the CNS entry points of afferents from the wings and halteres, and the location of ascending projections from some of these afferents. Scale bar, 100 µm. (**G**) Wing grooming bout rate with optogentic activation of neurons targeted by R30B01-AD ∩ R31H10-DBD. Data are displayed as described for *Figure 3C*. Asterisks: p<0.0001.

DOI: https://doi.org/10.7554/eLife.28804.016

The following figure supplement is available for figure 4:

**Figure supplement 1.** GAL4 lines that express in wing sensory neurons.

DOI: https://doi.org/10.7554/eLife.28804.017

movements are activated in parallel, and are thus selected in a hierarchically determined competition through suppression.

## Sensory neurons involved in grooming behavior

One aim of this work was to identify sensory neurons that can induce grooming behavior. The bristles are canonically thought to be involved in insect grooming based on evidence that their tactile stimulation on different body parts induces site directed grooming responses (*Vandervorst and Ghysen, 1980*; *Corfas and Dudai, 1989*; *Page et al., 2004*). Here, we provide evidence that direct activation of the bristle mechanosensory neurons can elicit grooming. We identify the fruit fly inter-ommatidial bristle mechanosensory neurons based on their anatomical similarity to those of the

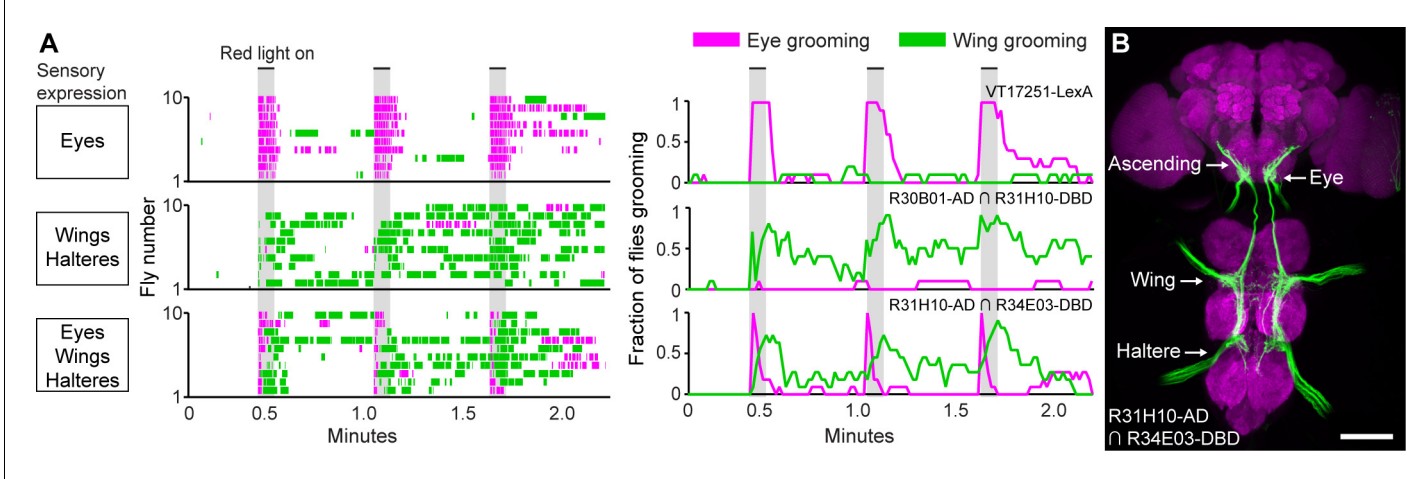

**Figure 5.** Simultaneous excitation of eye and wing/haltere sensory neurons produces sequential grooming. (A) Ethograms (left) and histograms (right) showing eye grooming (magenta) or wing grooming (green) elicited with red light-activated CsChrimson expressed in different transgenic lines. The lines express in sensory neurons on the eyes (VT17251-LexA (top row)), wings and halteres (R30B01-AD ∩ R31H10-DBD (middle row)), or eyes, wings, and halteres (R31H10-AD ∩ R34E03-DBD (bottom row)). Data is plotted as described in *Figure 2*. See *Video 6*, *Video 7*, and *Video 8* for representative examples. (B) GFP expression pattern of R31H10-AD ∩ R34E03-DBD in the CNS. Image shows a maximum intensity projection of a co-stain with anti-GFP (green) and anti-Bruchpilot (magenta). Arrows indicate the body part each sensory projection is from, and the location of ascending projections from the wings and halteres. Scale bars, 100 μm.

DOI: https://doi.org/10.7554/eLife.28804.018

The following figure supplement is available for figure 5:

**Figure supplement 1.** Simultaneous excitation of eye and wing/haltere sensory neurons produces sequential grooming.

DOI: https://doi.org/10.7554/eLife.28804.019

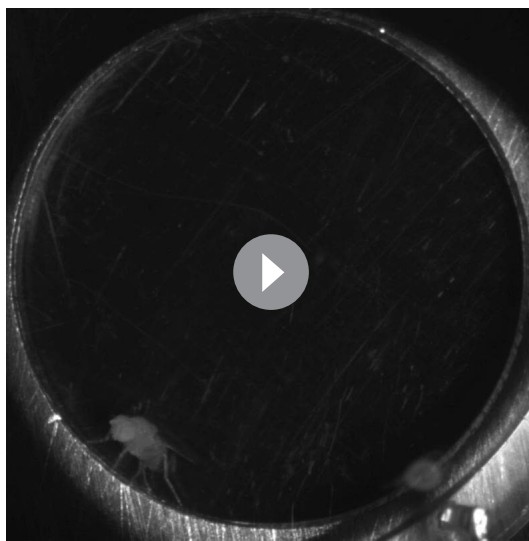

**Video 6.** Grooming in response to the optogenetic activation of eye bristle mechanosensory neurons. CsChrimson was expressed in eye bristle mechanosensory neurons using VT17251-LexA. The infrared light in the bottom right corner indicates when the red light was on to activate the sensory neurons.

DOI: https://doi.org/10.7554/eLife.28804.020

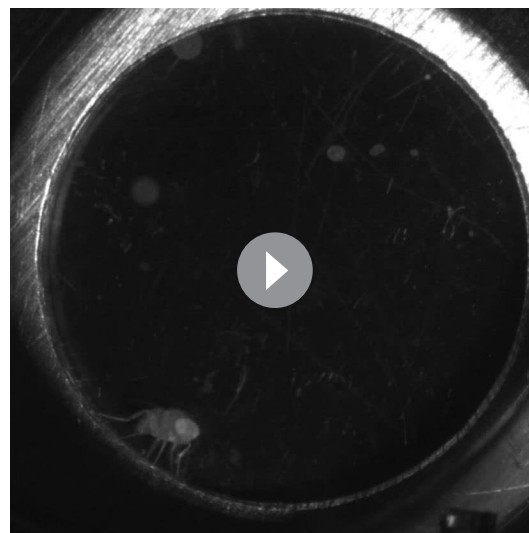

**Video 7.** Grooming in response to the optogenetic activation of wing/haltere sensory neurons. CsChrimson was expressed in wing and haltere sensory neurons using R30B01-AD ∩ R31H10-DBD. The infrared light in the bottom right corner indicates when the red light was on to activate the sensory neurons.

DOI: https://doi.org/10.7554/eLife.28804.021

praying mantis and cricket (*Honegger et al., 1979*; *Zack and Bacon, 1981*). Next, we use a transgenic driver line that expresses in these neurons to show that their optogenetic activation elicits eye grooming. We also identify different spGAL4 lines that express in neurons whose activation elicits wing grooming. However, these lines express both in bristle mechanosensory neurons and campaniform sensilla, raising the question of whether one or both sensory types are involved. Given the wealth of data implicating the bristles in grooming (*Tuthill and Wilson, 2016*), the parsimonious explanation is that the wing bristle mechanosensory neurons are involved. However, there is also a precedent for the involvement of non-bristle mechanosensory neurons such as the campaniform sensilla. For example, we previously showed that Johnston's Organ chordotonal neurons can detect displacements of the antennae to induce antennal grooming (*Hampel et al., 2015*), and others have shown that gustatory neurons on the wing can detect different chemicals to trigger grooming (*Yanagawa et al., 2014*). Therefore, further work is required to resolve which sensory neurons are involved in wing grooming.

One outstanding question is whether the sensory neurons have a direct role in establishing hierarchical suppression. We previously proposed two mechanisms of hierarchical suppression (*Seeds et al., 2014*). One is that unidirectional inhibitory connections between the movements drive suppression, a mechanism not likely to involve the sensory neurons. The other is that differences in sensitivity to dust across the body establish a gradient of sensory drives among the grooming movements, leading to suppression through winner-take-all competition. One way that sensitivity differences could be established is through differing numbers of receptors on each body part. For example, if we assume that the bristle mechanosensory neurons on the different body parts detect dust to elicit grooming (which remains to be shown), a comparison of bristle numbers on different body parts gives mixed support for this hypothesis. There are 600, 221, and 235 bristles reported to be on the eyes, wings, and notum respectively (*Hartenstein and Posakony, 1989*; *Cadigan et al., 2002*). The eyes are the highest priority part to be groomed, and have 2.7 times more bristles than the wings, which is consistent with the suppression hierarchy. In contrast, the lowest priority body part is the notum, which has more bristles than the wings, arguing against the hypothesis. Furthermore, given that other sensory neuron types elicit grooming (e.g. chordotonal and gustatory neurons), there may be multiple ways of detecting dust (*Yanagawa et al., 2014*; *Hampel et al., 2015*). Alternatively, hierarchical suppression could be established at the level of sensory neurons by regulating their output through presynaptic inhibition (*Blagburn and Sattelle, 1987*; *Burrows and Matheson, 1994*; *Clarac and Cattaert, 1996*; *Rudomin and Schmidt, 1999*). For example, the feeding behavior of the medicinal leech causes presynaptic inhibition of mechanosensory neurons, which suppresses touch-induced behavioral responses (*Gaudry and Kristan, 2009*). Future experiments will test such hypotheses about whether hierarchical suppression is established at the level of sensory neurons.

## Using optogenetic control of sensory neurons to further probe the neural mechanisms of hierarchical suppression

The use of optogenetics to activate sensory neurons reveals new insights into the neural mechanisms that drive grooming behavior. One striking example is evidence of a persistent trace of those sensory neurons that were optogenetically activated (next Discussion section). Other data in *Figure 5A* may show evidence of a prediction of the model of hierarchical suppression that later movements in the sequence can suppress earlier ones. A comparison of the eye grooming elicited by activating eye bristle mechanosensory neurons alone, versus co-activating eye and wing/haltere sensory neurons, reveals an earlier termination of eye grooming and the occurrence of wing grooming (compare *Figure 5A*, top and bottom panels). One explanation for this could be that wing grooming suppresses eye grooming. Our model indicates that grooming movements compete for output through a winner-take-all mechanism that selects the movement with the highest 'activation level' and suppresses others (*Seeds et al., 2014*). Activation levels are set by the relative amounts of dust on the body parts, and by a hierarchical weighting mechanism across the different movements. When more than one body part is covered in dust, the first movement is selected because it has the highest activity level. The removal of dust from that part causes the activity level to drop below that of the next movement that is consequently selected by suppressing the previous movement. Thus, the suppression of an earlier movement (e.g. eye grooming) by a later movement (e.g. wing grooming) is predicted by the model, providing a plausible explanation for the result.

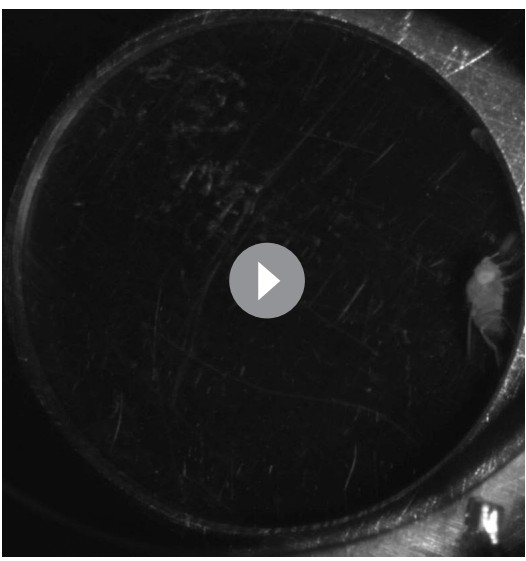

**Video 8.** Grooming in response to the simultaneous optogenetic activation of eye bristle mechanosensory and wing/haltere sensory neurons. CsChrimson was expressed in eye, wing, and haltere sensory neurons using R31H10-AD ⋂ R34E03-DBD. The infrared light in the bottom right corner shows when the red light was on to activate the sensory neurons.
DOI: https://doi.org/10.7554/eLife.28804.022

Although suppression of eye by wing grooming is predicted by the model, it is unclear why the flies transition to wing grooming during the optogenetic stimulus (*Figure 5A*, bottom panels). The continual optogenetic stimulation period might be likened to a hypothetical scenario where the fly could not remove dust from its eyes. Therefore, eye grooming might be expected to continue throughout the stimulation period, which would suppress wing grooming. The unexpected transition from eye to wing grooming during the stimulation period could occur if the activation level of eye grooming drops below wing grooming. One possible explanation is based on observations that continual tactile stimulation of bristle mechanosensory neurons causes habituation and loss of grooming responses (*Corfas and Dudai, 1989*). Similarly, optogenetic stimulus-dependent habituation might decrease the sensory drive to the point that wing grooming suppresses eye grooming, but not enough to abolish eye grooming when it is activated alone (as seen in the top panel of *Figure 5*). Another possibility is that the activation level of posterior grooming increases to the point that it surpasses eye grooming. There is evidence that grooming responses can temporally sum consecutive sensory stimuli, indicating a mechanism that can store and increase excitability to elicit a grooming response (*Sherrington, 1906*; *Stein, 2005*; *Guzulaitis et al., 2013*). Experiments that address these and other possible mechanisms will provide new insights into the neural mechanisms that drive the grooming sequence.

Future studies will address how changing the relative activation levels between grooming movements results in a shift of suppression such that later movements suppress earlier ones. This could be tested by extending the fiber optic-directed optogenetic approach used in this study (*Figure 1H*), such that two different body part sensory populations are differentially stimulated with adjustable relative levels of optogenetic illumination. This would allow for an assessment of whether reduced optogenetic light power on the eyes, and increased light power to the wings results in prioritized grooming of the posterior body. In effect, changing the illumination levels on the different body parts could mimic the loss of dust as flies clean the different parts of their bodies.

## Persistent neural activity within grooming neural circuits

Emerging behavioral evidence indicates that neural circuits controlling *Drosophila* grooming movements have mechanism(s) for maintaining excitability. This was originally proposed from a study identifying a mechanosensory circuit that elicits persistent grooming of the antennae (*Hampel et al., 2015*). That is, neurons within this circuit elicit grooming that continues for tens of seconds beyond their optogenetic activation. Work presented here reveals that activation of wing sensory neurons similarly elicits persistent grooming. Interestingly, grooming responses that outlast their stimulus have also been described in vertebrates, suggesting that persistence is a common feature of grooming (*Sherrington, 1906*; *Stein, 2005*). Despite the prevalence of persistent grooming, its biological function remains unclear. One possibility is that persistence prevents unnecessary switches between behaviors (*Redgrave et al., 1999*); for example swimming responses can last beyond the initial stimulus so that an animal can safely avoid a predator. In the case of grooming, persistence may ensure that a dirty body part is thoroughly cleaned before switching to another behavior.

We also infer the maintenance of excitability within grooming neural circuits from the observation that brief activation of sensory neurons across the body elicits a grooming sequence. That is, flies

groom their heads and then transition to their posterior bodies, even during the period after the red light has turned off. This indicates that flies maintain a persistent trace of which body parts are stimulated to elicit a delayed and sequential grooming response. We postulate that this occurs when the simultaneous stimulation of sensory neurons across the body activates each grooming movement in parallel. Eye grooming occurs first by suppressing grooming movements occurring later (*Seeds et al., 2014*), however the circuitry for each later movement remains active without requiring further sensory input. The next movement is then elicited via this persistent neural activity once suppression from eye grooming ceases. If this is the case, it raises the question of how the previous movement terminates to allow the next movement to proceed. Further, it is unclear how circuits that drive later grooming movements retain neural excitability. Such acquisition and maintenance of excitability is reminiscent of a previously described feature of grooming called *temporal summation*, whereby successive subthreshold stimuli are summed to elicit grooming (*Sherrington, 1906*; *Stein, 2005*; *Guzulaitis et al., 2013*). Thus, both temporal summation and the grooming sequence observed here point to a mechanism within the grooming neural circuitry that maintains a persistent trace of the sensory stimulus.

How does a mechanism that maintains excitability within the grooming neural circuitry affect our previously proposed model of grooming behavior? Our previous model indicated that constant stimulation is necessary for each grooming movement to be active (*Seeds et al., 2014*). That is, dust on a body part provides a constant drive to groom that is lessened through its removal. Indeed, a computational model where the movements are driven entirely by the presence of dust produces grooming that resembles dust-induced grooming. This indicates that the model well describes grooming that occurs over relatively long time scales (~30 min). However, based on observations that grooming persists after a brief stimulus, we now propose that the circuitry contains a neural mechanism that allows grooming movements to remain active on shorter time scales (tens of seconds). The ability to identify and manipulate the sensory neurons that elicit grooming movements and their downstream circuits now enable experiments to determine how persistent neural excitability is acquired and maintained.

## Materials and methods

### Fly stocks and rearing conditions

The GAL4 lines used in this study were produced by Gerald Rubin's lab at Janelia Research Campus and are available from the Bloomington *Drosophila* stock center (*Jenett et al., 2012*). The lines were identified in a screen for those that expressed GAL4 in neurons whose activation could elicit grooming behavior (*Seeds et al., 2014*). In this work, we screened through the images of the CNS expression patterns of these GAL4 lines (*Jenett et al., 2012*), searching for those with expression in afferents from each of the different body parts (*Figure 1—figure supplement 1A–D*). These lines were selected for detailed behavioral and anatomical analysis as described in the results section. The control used for the GAL4 lines was BDPGAL4U, which contains the vector backbone used to generate each GAL4 line (including GAL4), but lacks any enhancer to drive GAL4 expression (*Seeds et al., 2014*). The Split GAL4 stocks were produced by Gerald Rubin's lab according to previously described methods (*Pfeiffer et al., 2010*). VT17251-LexA was a gift from the lab of Barry Dickson. Controls for the Split GAL4 stocks were produced in the same way as BDPGAL4U, but each spGAL4 half was used in place of GAL4 (*Hampel et al., 2015*).

Transgenic flies carrying the following UAS drivers were from the following citations: *UAS-dTrpA1* (*Hamada et al., 2008*), 20x*UAS-mCD8::GFP* (pJFRC7) (*Pfeiffer et al., 2010*), 13xLexAop-myr::GFP (pJFRC19) (*Pfeiffer et al., 2010*), *UAS-Channelrhodopsin-2* (*Hwang et al., 2007*), and 20x*UAS-CsChrimson* (attP18) (*Klapoetke et al., 2014*). GAL4, spGAL4, and LexA lines were crossed to their respective UAS or LexAop drivers, and the progeny were reared on cornmeal and molasses food at 21°C and 50% relative humidity using a 16/8 hr light/dark cycle. For optogenetic experiments using CsChrimson or ChR2, flies were reared in the dark on food containing 0.4 mM all-*trans*-retinal. All experiments were done with five to eight day old males.

## Channelrhodopsin-mediated activation of sensory neurons using decapitated flies

Two different regions of the bodies of decapitated flies were illuminated to locally activate sensory neurons expressing Channelrhodopsin (ChR2). ChR2 was used for these experiments because the intensity of the blue light used for its activation is attenuated by the cuticle of the fly (*Inagaki et al., 2014*). Therefore, blue light-elicited grooming would likely be caused by activation of the sensory neurons on the illuminated region of the body surface rather than activation of afferents inside the fly that project to the CNS from other body parts. For decapitations, flies were cold-anesthetized, decapitated using a standard razor blade, and allowed to recover for 10–20 min. The flies were positioned on a slide for the experiment using a fine paint brush. A 473 nm blue light LED (Nichia Corp, Tokushima, Japan) was attached to an optical fiber (1 mm in diameter) to direct light to a specific region on the fly. The optical fiber was held approximately 1 mm from the target body region to deliver a blue light stimulus with a luminance of 0.075 mW/mm$^2$. The light was directed towards the dorsal posterior region, away from the anterior body, to activate sensory neurons primarily on the wings (*Figure 1—figure supplement 2A*). Alternatively, the light was directed towards the dorsal anterior region, away from the posterior body, to activate sensory neurons primarily on the notum (*Figure 1—figure supplement 2B*). The LED stimulus was controlled using a Grass SD9 stimulator (Astro-Med Inc., Warwick, RI) that delivered 10 Hz pulses that were 20 milliseconds in duration, with 8-millisecond delays between pulses. Each fly was subjected to stimulation on each body region in random order; however, in some cases the flies would jump during the experiment and could not be used further. A grooming response to the illuminated body region within a ten-second time frame was scored as a positive response. The fraction of flies that responded was plotted. The number of trials for each dorsal body region for each line were: R52A06-GAL4, anterior (n = 100), posterior (n = 100); R30B01-GAL4, anterior (n = 40), posterior (n = 61); R81E10-GAL4, anterior (n = 86), posterior (n = 89). Statistical significance was addressed using Chi-Square tests and Bonferroni correction.

## CsChrimson-mediated activation of sensory neurons using freely moving flies

The camera and behavioral setups used for recording freely moving flies with optogentic activation were described previously (*Seeds et al., 2014*; *Hampel et al., 2015*). Flies were cold anesthetized, loaded into behavioral chambers, and allowed to recover for at least ten minutes. R52A06-, R30B01-, and R81E10-GAL4 were used to express the light-gated channel CsChrimson. For this experiment, we used CsChrimson for optogenetic activation rather than ChR. In contrast to the blue light used for ChR, the red light used for gating CsChrimson readily penetrates the fly cuticle (*Inagaki et al., 2014*), allowing for uniform activation of sensory neurons across the body, regardless of the flies' orientation. Our initial experiments using optogenetic activation of the neurons targeted by these GAL4 lines revealed that high levels of red light activation caused defects in motor coordination. This was likely caused by the strong activation of sensory neurons across the body, some of which are known to be involved in proprioception (e.g. femoral chordotonal organs). Therefore, it was necessary to reduce the red-light power to the point where it elicited grooming without causing coordination defects. The light power that met these requirements for each GAL4 line are: R52A06-GAL4 (0.066 mW/mm$^2$), R30B01-GAL4 (0.066 mW/mm$^2$), and R81E10-GAL4 (0.077 mW/mm$^2$). The light power used for each LexA and spGAL4 line was: VT17251-LexA (0.382 mW/mm$^2$), R30B01-AD ∩ R31H10-DBD and R31H10-AD ∩ R34E03-DBD (0.135 mW/mm$^2$). The red light frequency was 5 Hz (0.1 s on/off) for 5 s, followed by 30 or 60 s intervals where the red light was off. The experiment consisted of a total of three photostimulation periods with 30 or 60 s intervals between each stimulation. The experiment was recorded for manual annotation of the grooming movements performed.

The recorded grooming movements of flies were manually annotated as described previously (*Seeds et al., 2014*). For the ethogram and histogram plots, the different head grooming movements (e.g. eye, antennal, and proboscis grooming) were binned (1 s time bins) and plotted as head grooming. Similarly, all movements that were directed towards the body (e.g. abdomen, wings, notum) were binned and plotted as posterior body grooming. Statistical analysis and display of the data in *Figure 3* and *Figure 4* were previously described (*Hampel et al., 2015*). The boxplots shown in *Figure 2—figure supplement 2* represent the percent time that flies spent grooming within the intervals of the stimulation regime shown in *Figure 2* (i.e. Pre-stimulation, red light on, and post-

stimulation rest periods 1–3). Statistical analysis was done using the Friedman test followed by post-hoc Wilcoxon signed rank test for pairwise comparisons with Bonferroni-Holm correction for multiple comparisons. *Figure 5—figure supplement 1A* was plotted as the mean fraction of flies performing head or posterior grooming in each frame across the three different stimuli. The envelope was calculated as the standard error of the mean. For the bar plot shown in *Figure 5—figure supplement 1A*, the average time to groom after stimulus onset was defined as the first frame where each fly performed either head or posterior grooming after the stimulus onset. Statistical significance was assessed using a Mann-Whitney U test.

## Analysis of CNS and PNS expression patterns

Dissection and staining of the CNS was performed using a previously reported protocol (*Hampel et al., 2011*). The head stain shown in *Figure 3A* was done as follows. Fine scissors were used to cut off part of the proboscis and part of the eyes to improve antibody penetration. Heads were fixed in phosphate buffered saline (PBS) containing 2% paraformaldehyde and 0.1% Triton for 2 hr at 4°C, and stained with primary antibodies: rabbit anti-GFP (1:500, Thermo Fisher Scientific, Waltham, MA, #A11122) and mouse anti-nc82 (1:50, Developmental Studies Hybridoma Bank, University of Iowa) followed by secondary antibodies: goat anti-rabbit DyLight 594 (Thermo Fisher Scientific #35560) and goat anti-mouse DyLight 633 (Thermo Fisher Scientific #35512), with Calcofluor White to stain the cuticle (a few grains in 300 µl volume, Sigma #F3543). Images were collected using a Zeiss LSM710 confocal microscope using a Plan-Apochromat 20x/0.8 M27 objective (Carl Zeiss Corporation, Oberkochen, Germany).

Dissection of the different body parts and imaging of the PNS expression patterns of the different GAL4, LexA, and Split GAL4 lines were performed as follows. The lines were crossed to 20x*UAS-mCD8::GFP* (JFRC7) or 13xLexAop-myr::GFP (pJFRC19). The progeny were anesthetized using $CO_2$, decapitated, dipped in 70% ethanol, transferred to PBS, and each body part was dissected as described below. The unfixed body parts were imaged immediately in PBS or Vectashield (Vector Laboratories, Burlingame, CA). We used both PBS and Vectashield and did not notice a difference in the cell morphology or expression pattern when using either reagent. The use of Vectashield had the advantage of resulting in fewer air bubbles between the coverslip and sample.

*Head:* Flies were decapitated using a standard razor blade. Heads were then placed 'face up' on a slide in a small well that was made by stacking six reinforcement labels (Avery Dennison Corporation, Brea, CA) and filled with PBS or Vectashield. A cover slip was then placed over the well. *Abdomen:* The abdomen was severed from the rest of the body just posterior to the scutellum. Abdomens were then placed on a slide in a well created as described above. The abdomens were placed either ventral or dorsal side up so that each side could be imaged. *Notum:* A scalpel was used to slice longitudinally between the legs and the dorsal side of the notum. The notum was imaged in the same well preparation described above. *Wing:* A scalpel was used to remove the left wing from the body of the fly. To ensure that the entire wing was obtained, part of the body wall was also cut with the wing. The wing was then placed on a drop of Vectashield and then covered with a coverslip. *Leg:* The left prothoracic leg was dissected in the same way as the wing. All body parts were imaged using a Zeiss 710 confocal microscope using 10x and 20x air objectives. Native GFP fluorescence was imaged using an excitation wavelength of 488 nm, whereas autofluorescence from cuticle was imaged using 568 nm. Body parts from at least three flies were imaged from separate crosses on different days. In some cases, the body parts were imaged at 20x and then stitched using a FIJI plugin (*Preibisch et al., 2009*).

The different sensory neuron types on each body part were classified based on previous descriptions (*Ghysen, 1980*; *Cole and Palka, 1982*; *Dickinson and Palka, 1987*; *Murphey et al., 1989*; *Smith and Shepherd, 1996*). The numbers of campaniform sensilla and mechanosensory bristle neurons on the wings were previously counted (*Cole and Palka, 1982*; *Dickinson and Palka, 1987*; *Hartenstein and Posakony, 1989*). Proximal campaniform sensilla described in this work include ANWP, Teg, d.Rad.A, d.Rad.B, d.Rad.C, d.Rad.D, d.Rad.E, d.HCV, v.Rad.A, v.Rad.B, v.Rad.C, v. HCV, and vL.III. Distal campaniform sensilla described in this work include GSR, p.TSM, d.TSM, L3-V, ACV, L3-1, L3-2, and L3-3. We classified neurons on the wings as bristle mechanosensory rather than chemosensory given that their dendrites appear to terminate at the base of the bristle rather than projecting to the bristle tip.

## Acknowledgements

We thank: David Shepherd for sharing his knowledge of sensory systems; Gerry Rubin and his lab for generating the spGAL4 stocks used in this work; Aljoscha Nern for advice on identifying eye bristle CNS expression patterns; Eric Hoopfer for Matlab code for producing histogram plots, time binned box plots, and comments on the manuscript; Phuong Chung for help imaging body parts; Igor Siwanowicz for suggestions on head staining methods; and Jonathan Blagburn for comments on the manuscript. Funding for this work was provided by the Howard Hughes Medical Institute, the COBRE Center for Neuroplasticity, NIH NIGMS GM103642, and NIH MD007600.

## Additional information

### Funding

| Funder | Grant reference number | Author |
| --- | --- | --- |
| Howard Hughes Medical Institute | | Julie H Simpson |
| National Institutes of Health | GM103642 | Andrew M Seeds |
| National Institutes of Health | MD007600 | Andrew M Seeds |

The funders had no role in study design, data collection and interpretation, or the decision to submit the work for publication.

### Author contributions

Stefanie Hampel, Conceptualization, Data curation, Formal analysis, Investigation, Methodology, Writing—review and editing; Claire E McKellar, Resources, Investigation, Methodology, Writing—review and editing; Julie H Simpson, Conceptualization, Resources, Funding acquisition, Writing—review and editing; Andrew M Seeds, Conceptualization, Data curation, Formal analysis, Funding acquisition, Investigation, Methodology, Writing—original draft, Writing—review and editing

### Author ORCIDs

Andrew M Seeds http://orcid.org/0000-0002-4932-6496

### Decision letter and Author response

Decision letter https://doi.org/10.7554/eLife.28804.023
Author response https://doi.org/10.7554/eLife.28804.024

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
