## [Decision Letter]

Thank you for submitting your article "Simultaneous activation of parallel sensory pathways promotes a grooming sequence in *Drosophila*" for consideration by *eLife*. Your article has been reviewed by three peer reviewers, one of whom, Ronald L Calabrese (Reviewer #1), is a member of our Board of Reviewing Editors, and the evaluation has been overseen by K VijayRaghavan as the Senior Editor.

The reviewers have discussed the reviews with one another and the Reviewing Editor has drafted this decision to help you prepare a revised submission.

Summary:

This is an interesting advance on a seminal paper in *eLife* in 2014, which established the grooming behavior of *Drosophila* as a very productive one for the study of sequential behaviors at the cellular level, using all the genetic tricks available. In that paper the authors proposed a model of hierarchical suppression where simultaneous activation of sensory neurons on different body parts (dusting the whole fly) elicits a prioritized grooming response, roughly anterior to posterior, with eye grooming dominant over wing grooming, for example. In this advance, the authors use Gal4 and spGal4 drivers to begin the identification of the sensory neurons involved in grooming. Whole body short duration optogenetic stimulation of using general Gal4 drivers for sensory neurons gives rise to sequencing with eye grooming not giving way to posterior body grooming until the activating light is turned off, but light stimulation limited to head or body give rise to eye grooming or body grooming respectively. Thus a persistent state of activation of posterior body grooming is revealed and the model is supported. They identify with spGal4 drivers and optogenetically activate specific sensory neurons that elicit different specific grooming movements, eye grooming and wing grooming. The goal is to activate them optogenetically together and test their effects on grooming behavior and in particular on its sequencing. The ultimate experiments, once limited identified populations of eye and the wing sensory neurons are under optogenetic control, was to simultaneously activate them and show that eye input remains dominant but sequencing persists. This is a significant advance in technique and it provides strong evidence for the hierarchical suppression model, which is of wide interest. The tools and reagents are now in place to explore directly how suppression is accomplished by sensory neurons. The work is carefully done with appropriate controls and characterization of Gal4 and spGal4 lines. The data is analyzed in an appropriate manner. The lines characterized will be very useful for fly workers who wish to pursue experiments in which limited classes of sensory neurons are to be optogenetically activated. The work is of interest to those who study the neuronal bases of natural behaviors and behavior sequences.

Essential revisions:

1) The experiment of Figure 5 seem to show that stimulation of wing/halters sensory neurons has an inhibitory effect on head grooming; eye grooming is very limited in time when the wings/halters are stimulated together with the eyes compared to the eyes alone. All three reviewers commented on this seeming interaction in different ways. The authors should explain what they think is going on here and any experiments that balance the optogenetic stimulation of eye vs wing/halters sensory neurons differently.

2) Experiments could, in principle, be done using the activation of mechanosensors in the abdomen, playing them against activation of the head and/or thoracic mechanosensors. The authors should discuss whether there is a reason that such experiments were not done.

*Reviewer #1:*

1) Paragraph two subsection “Competition between eye and wing sensory neurons elicits prioritized grooming” and Figure 5 am not totally convinced by the authors interpretation 'We also found evidence of suppression by eye grooming, as ongoing grooming of the wings terminated and all flies groomed their eyes with each red light stimulus…". What I see in the bottom panels of Figure 5 is that posterior body grooming begins during the light pulse and that eye grooming terminates rapidly. This is in contrast to the top panes where eye grooming is much more long lasting; outlasting the pulse of light even. There appears to be inhibition of the eye grooming by the body grooming or at least bye the wing-halteres stimulus. Can the authors expand on what is happening here?

2) Paragraph three of subsection “Simultaneous excitation of sensory neurons across the body induces a grooming sequence” and Figure 2: Have you tested shorter ISIs to determine whether the light stimulus can suppress body grooming when it is most robust?

*Reviewer #2:*

This is an excellent follow-up study to this group's 2014 *eLife* paper that proposes a suppression hierarchy that drives sequential grooming behavior by adult fruit flies. This study nicely tests the basic assumption of the model proposed in the previous study-that activation of mechanosensory neurons all over the fly's body activates pattern generators that arouses grooming behavior to all parts of the body, with the location of the grooming sequenced in an anterior-to-posterior pattern. The experiments are well presented, appropriate, and the results are convincing. The conclusions are fully justified and nicely qualified where appropriate.

Their basic technique (using combinations of GAL-4 lines to target mechanosensory neurons in different areas of the body, then activating them optogentically) provides a strong test of the hypothesized mechanism. The results are convincingly in line with the hypothesis and, additionally, provide the added information that even short-lasting activation of a set of mechanosensors produces a long-lasting bout of grooming the appropriate area, a finding that should lead to interesting extensions of this work in the future.

Two possible extensions of the research that should be addressed:

1) The same kinds of experiments could, in principle, be done using the activation of mechanosensors in the abdomen, playing them against activation of the head and/or thoracic mechanosensors. The authors should discuss whether there is a reason that such experiments were not done.

2) Have the authors tried weakly stimulating head mechanosensors and strongly stimulating the thoracic ones? Such an experiment would test whether there is a weak back-to-front inhibition of grooming behavior in addition to the clearly strong front-to-back inhibition. If they have such data, it would be good to show it. If not, they might mention this as a possible future experiment.

*Reviewer #3:*

This paper represents an important addition to the previous studies by this group looking at the emergence of grooming sequences in *Drosophila*. Overall, I find the experimental data compelling, the writing is clear, and I have no major experimental concerns.

---

## [Author Response]

Essential revisions:1) The experiment of Figure 5 seem to show that stimulation of wing/halters sensory neurons has an inhibitory effect on head grooming; eye grooming is very limited in time when the wings/halters are stimulated together with the eyes compared to the eyes alone. All three reviewers commented on this seeming interaction in different ways. The authors should explain what they think is going on here and any experiments that balance the optogenetic stimulation of eye vs wing/halters sensory neurons differently.

We agree with the reviewers that the result shown in Figure 5 could be accounted for by suppression of eye by wing grooming. Based on the unanimous response by the reviewers on this point we now address it in the Results and a new Discussion section. We explain how these results are consistent with our model. To achieve mutual exclusivity, each grooming movement can suppress others, but in accordance with the hierarchy, anterior movements tend to suppress posterior ones more strongly.

In the Figure 5 Results section, we added the following to the last paragraph:

“Interestingly, the transition of eye to wing grooming occurred within the stimulus period, suggesting that eye grooming became inhibited by wing grooming during stimulus. This may reveal a prediction of the model of hierarchical suppression that later movements in the sequence can suppress earlier ones (see Discussion). However, the prioritized suppression came from eye grooming, as any ongoing grooming of the wings terminated and all flies groomed their eyes with each red light stimulus (Figure 5, bottom, Figure 5—figure supplement 1).”

The following text has been added as a new Discussion section. This section also addresses the requested extension of reviewer 2: “Have the authors tried weakly stimulating head mechanosensors and strongly stimulating the thoracic ones?”

*“*Using optogenetic control of sensory neurons to further probe the neural mechanisms of hierarchical suppression. The use of optogenetics to activate sensory neurons reveals new insights into the neural mechanisms that drive grooming behavior. […] In effect, changing the illumination levels on the different body parts could mimic the loss of dust as flies clean the different parts of their bodies.”

2) Experiments could, in principle, be done using the activation of mechanosensors in the abdomen, playing them against activation of the head and/or thoracic mechanosensors. The authors should discuss whether there is a reason that such experiments were not done.

We agree with the reviewer that this would be a very interesting set of experiments to do. However, we first need to identify transgenic driver lines (e.g. Split GAL4 lines) that specifically target two populations of neurons on the abdomen and head or thorax. This is a laborious, and often unsuccessful process that involves screening though many different combinations of Split GAL4 halves (i.e. AD and DBDs), searching for those that express exclusively in both populations of sensory neurons. It has been particularly difficult to identify lines that express in bristle mechanosensory neurons on specific body parts, which are likely the primary sensory neurons whose activation elicits abdominal and thoracic grooming. The main problem seems to be that the lines often express in mechanosensory bristles on several different body parts. The eye bristle driver that we identified and described in this manuscript seems to be an exception to that rule. Although we had success in identifying a Split GAL4 driver line that expresses in eye and wing/haltere mechanosensory neurons, we have yet to identify a driver that expresses exclusively in the abdomen and head or thorax. This is the reason why we have not tried this experiment yet.

Reviewer #1:1) Paragraph two subsection “Competition between eye and wing sensory neurons elicits prioritized grooming” and Figure 5 am not totally convinced by the authors interpretation 'We also found evidence of suppression by eye grooming, as ongoing grooming of the wings terminated and all flies groomed their eyes with each red light stimulus…". What I see in the bottom panels of Figure 5 is that posterior body grooming begins during the light pulse and that eye grooming terminates rapidly. This is in contrast to the top panes where eye grooming is much more long lasting; outlasting the pulse of light even. There appears to be inhibition of the eye grooming by the body grooming or at least bye the wing-halteres stimulus. Can the authors expand on what is happening here?

We addressed this reviewer’s concern in the essential revisions section above.

2) Paragraph three of subsection “Simultaneous excitation of sensory neurons across the body induces a grooming sequence” and Figure 2: Have you tested shorter ISIs to determine whether the light stimulus can suppress body grooming when it is most robust?

This is an interesting question. The model of hierarchical suppression predicts that optogenetic activation of the sensory neurons at shorter interstimulus intervals than the 60 seconds shown in Figure 2 would similarly cause the flies to cease grooming their posterior bodies and transition to grooming their heads. As the reviewer asked, we repeated the experiment using a shorter interstimulus interval (30 seconds). To ensure that the optogenetic stimulus was delivered, to quote the reviewer, “when it [posterior grooming] is most robust”, we performed several trials of this experiment, but only plotted the trials where the flies were engaged in posterior grooming at the time of the next stimulus. In all cases, the animals ceased grooming their posterior bodies and groomed their heads. We added an additional figure supplement that shows the results of this experiment (Figure 2—figure supplement 4). We also added text in the Results section of Figure 2: “This termination of posterior grooming was still observed when we shortened the time between light stimuli and only examined trials where flies were grooming at the moment the next stimulus was delivered (Figure 2—figure supplement 4).” This new experiment strengthens our original argument that activating head sensory neurons can initiate anterior grooming and suppress ongoing posterior grooming.

Reviewer #2:[…] Two possible extensions of the research that should be addressed:1) The same kinds of experiments could, in principle, be done using the activation of mechanosensors in the abdomen, playing them against activation of the head and/or thoracic mechanosensors. The authors should discuss whether there is a reason that such experiments were not done.

We addressed this reviewer’s concern in the essential revisions section above.

2) Have the authors tried weakly stimulating head mechanosensors and strongly stimulating the thoracic ones? Such an experiment would test whether there is a weak back-to-front inhibition of grooming behavior in addition to the clearly strong front-to-back inhibition. If they have such data, it would be good to show it. If not, they might mention this as a possible future experiment.

We agree that this would be an excellent experiment to try. However, we do not yet have the necessary hardware to rigorously do this experiment. We are constructing an apparatus that uses two lasers to illuminate two body parts as the same time with different light powers. This will allow us to adjust the relative optogenetic light power on the two parts and test whether weak stimulation of the anterior body with strong stimulation of a posterior part results in the prioritization of posterior grooming and the suppression of anterior grooming.

We realized with this reviewer’s comment that this would be a good point to mention in the manuscript, and added a paragraph to the new Discussion section called: “Optogenetic control of sensory neurons to further probe the neural mechanisms of hierarchical suppression.” The entire Discussion section is also shown above. The paragraph that is directly relevant to this reviewer’s comment follows:

“Future studies will address how changing the relative activation levels between grooming movements results in a shift of suppression such that later movements suppress earlier ones. […] In effect, changing the illumination levels on the different body parts could mimic the loss of dust as flies clean the different parts of their bodies.”